# Cutaneous Metastases—Histological Particularities of Multifaceted Entities

**DOI:** 10.3390/dermatopathology12020014

**Published:** 2025-04-25

**Authors:** Andreea Cătălina Tinca, Bianca Andreea Lazar, Andreea Raluca Cozac-Szőke, Georgian Nicolae Radu, Simina Petra Simion, Diana Maria Chiorean, Irina Bianca Kosovski, Adrian Horațiu Sabău, Raluca Niculescu, Iuliu Gabriel Cocuz, Raluca-Diana Hagău, Emoke Andrea Szasz, Sabin Gligore Turdean, Ovidiu Simion Cotoi

**Affiliations:** 1Pathophysiology Department, George Emil Palade University of Medicine, Pharmacy, Science and Technology, 540142 Târgu-Mureș, Romaniaandreea-raluca.szoke@umfst.ro (A.R.C.-S.); diana.chiorean@umfst.ro (D.M.C.); bianca.kosovski@umfst.ro (I.B.K.); adrian-horatiu.sabau@umfst.ro (A.H.S.); raluca.niculescu@umfst.ro (R.N.); iuliu.cocuz@umfst.ro (I.G.C.); ovidiu.cotoi@umfst.ro (O.S.C.); 2Pathology Department, Mureș Clinical County Hospital, 540003 Târgu Mureș, Romania; george.radu098@gmail.com (G.N.R.); simionsimina70@yahoo.ro (S.P.S.); ralucahagau1995@gmail.com (R.-D.H.); emoke.szasz@umfst.ro (E.A.S.); sabin.turdean@umfst.ro (S.G.T.)

**Keywords:** cutaneous metastases, cancers, immunohistochemistry

## Abstract

Cutaneous metastases from internal organ cancers are diagnosed in approximately 0.2% of skin biopsies. This diagnosis can be the first sign of a previously undiagnosed malignancy with an internal organ origin. We conducted a retrospective study that included all cases of cutaneous metastases diagnosed in our hospital. A total of 25 patients were identified (14 females and 11 males). The average age of the patients included was 62.3. The most common primary cancer site was the lung for male patients, while for female patients it was the breast. In seven of our cases, cutaneous metastases were the first sign of an internal organ cancer. Common sites for cutaneous metastases in our study involved the anterior thoracic wall, the abdomen, and the scalp. Our study aims to highlight the importance of recognizing the histopathology of metastatic tumors and differentiating them from primary skin neoplasms. Immunohistochemistry is a mandatory tool for differential diagnosis in all cases, especially for patients who do not have a history of neoplasia.

## 1. Introduction

The skin is the largest organ of the human body and is important for protecting internal structures, temperature regulation, sensory information, and contributing to maintaining the overall homeostasis [1]. The skin also serves as a site for the manifestation of systemic conditions for which specific and non-specific histopathological characteristics can be identified [2,3,4].

Skin neoplasias, both benign and malignant, are quite commonly encountered. Some of the most frequent benign tumors of the skin are nevi, seborrheic keratoses, and papilloma. For the malignant category, basal cell carcinoma represents a majority, followed by more aggressive cancers such as squamous cell carcinoma and melanoma. Metastases are rare; less than 10% of the tumors can secondarily involve the skin. Secondary tumors from viscera account for approximately 0.2% of skin metastases. Such an event is correlated with poor prognosis. Often, distinguishing between primary skin carcinoma and metastatic carcinoma of various origins can be troublesome, requiring specific immunohistochemical markers, alongside clinical and paraclinical correlation [5,6,7,8].

Cutaneous metastases can be classified into three distinct groups, based mostly on the mechanism of invasion. Thus, tumoral cells can affect the skin by contiguity, by lymphatic, or by hematogenous spread [9,10]. Based on the primary tumor’s location, we have skin metastases originating from primary cutaneous malignant tumors, such as primary skin melanoma, squamous cell carcinoma, Merkel cell carcinoma, and adnexal carcinomas. These represent by far the largest group, with melanoma leading with 45%, according to some studies, followed by Merkel cell carcinoma with 25%. The second group includes metastases from internal malignancies, with breast cancer being seen most frequently (30% of the cases). The final group consists of cutaneous manifestations of systemic hematological neoplasms, such as leukemia and lymphomas [11].

While skin metastases are often a late manifestation of the disease, in some instances, they may be the first indication of internal malignancies. This can occur more often in lung, breast, renal, and ovarian cancers [12,13,14].

Histologically, metastatic lesions present significant heterogeneity, and in some cases, the classic features of the primary tumor may be absent. Immunohistochemistry is therefore an essential tool in establishing the diagnosis. The expression profiles of tumor cells vary depending on their origin, and antibody panels can guide the diagnostic process. For example, GATA3 in combination with ER, mammaglobin, and PR supports the breast origin; TTF1 and Napsin A suggest a pulmonary source; CK20 and CDX2 indicate a neoplasia of the lower gastrointestinal tract, CD10, alongside PAX8, typically points toward renal origins, and many more. The need for marker combinations and the patient’s clinical history must all be carefully considered. Despite its necessity and importance, immunohistochemistry interpretation is not always straightforward because expression patterns can vary significantly. This occurs frequently in poorly differentiated tumors, leading to diagnostic uncertainty and requiring integration of morphology, marker combinations, and clinical context [15,16,17,18].

The importance of recognizing cutaneous metastases lies not only in confirming the diagnosis of the metastatic disease but also in identifying a possible unknown primary tumor. While several case reports and small series exist in the literature, there are only a few cohort-based analyses.

The aim of this study is to evaluate the histological and immunohistochemical characteristics of cutaneous metastases from internal neoplasms diagnosed in our hospital. By doing so, we emphasize the diagnostic challenges and highlight the role of pathology in identifying and differentiating metastatic skin tumors from primary skin cancers.

## 2. Materials and Methods

This retrospective study included all cases of cutaneous metastases with visceral origin diagnosed between 2016 and 2024 in the Department of Pathology at a tertiary hospital in Romania (n = 25). Only histologically confirmed cases originating from internal organ neoplasms were included. Cases involving metastases from cutaneous tumors or other tumors with non-visceral origin were excluded from the analysis.

Brief clinical data (patient demographics, personal pathological history, and site of primary and secondary tumors) were reviewed. All cases were evaluated by at least two pathologists to confirm the diagnosis and immunophenotype of the tumors.

Tissue specimens were processed using standard histological techniques, including formalin fixation and paraffin embedding, followed by hematoxylin-eosin (HE) staining.

Immunohistochemical assessment was performed on 4 μm-thick sections prepared from formalin-fixed paraffin-embedded tissue, using an automated immunostainer (Benchmark GX, Ventana Medical Systems, Inc., Tucson, AZ, USA). All reagents and incubation times were chosen based on standard protocols that accompanied the antibody package inserts. Slides were developed using the OmniMap 3,3′ Diaminobenzidine (DAB) detection kit (Ventana Medical Systems, Inc.) and were counterstained with Hematoxylin.

The antibodiesused for the immunohistochemistry analysis are detailed in Table 1.

## 3. Results

Out of the 4820 skin resection specimens diagnosed by our department throughout an 8-year period, 25 cases (0.5%) were identified to fit the inclusion criteria (cutaneous metastases with visceral origin). The cases included samples from 14 women and 11 men. Seven patients presented cutaneous metastases as the first sign of a yet-undiagnosed internal malignancy.

The age of the patients at diagnosis varied significantly, ranging from 44 to 93, with a mean age of 62.3 ± 10.5 years. The mean age of presentation for females was 63.8 ± 11.3 years (ranging from 50 to 93 years old). The mean age for males was 60.7 ± 9.9 years (ranging from 44 to 80 years old) (Figure 1).

The chest was the most common site of metastases, and it was observed in eight cases. This region was mostly affected by breast and lung cancers. Other metastatic sites included the scalp, which was involved by metastases from urinary tract cancers, and the abdomen, which was frequently affected by gastrointestinal tract tumors.

The most common origin of the primary malignancy site overall, among our patients, was the lung (seven cases), followed by the breast (six cases), gastrointestinal tract (five cases), urinary tract (five cases), and reproductive tract (two cases), data shown in Table 2.

In women, breast cancer predominated as primary tumor (six patients), followed by malignancies of the gastrointestinal tract (two cases), reproductive system (two cases), urinary tract (one case), and lung (one case).

In men, the most common primary site was the lung (six patients), followed by urinary tract (four patients), and, lastly, by the gastrointestinal tract (three patients).

Breast carcinoma metastases were seen in the anterior chest wall (four cases) and the upper extremities (two cases). The most common histological subtype identified in these cases was NST carcinoma (four cases). The metastases were in the dermis and presented as groups and nests of cells with marked atypia. The tumor cells were positive for estrogen receptor (ER), GATA 3, E-cadherin, and CTK AE1/AE3 (Figure 2). We also diagnosed one metastatic mucinous breast carcinoma, characterized by groups, plaques, and clusters of cells surrounded by abundant mucin. The cells were positive for ER (Figure 3).

The last tumor was metastatic lobular carcinoma, where we identified cells distributed in rows, chords, and isolated, infiltrating the entire dermis. The cells were positive for CTK AE1/AE3, GATA3, ER, PR, and were negative for E-cadherin (Figure 4).

NST breast carcinoma represented two of the cases exhibiting cutaneous metastasis as the first sign of an internal neoplasm.

Cutaneous metastases of lung carcinoma were diagnosed in seven patients. Lung cancer was the most frequent primary neoplasm in which the cutaneous metastasis was the first sign of malignancy (five cases). In this cancer, the most common sites for skin metastases were the upper limbs (three cases), the anterior chest wall, and the shoulder (two cases).

Four cases were classified as squamous cell carcinoma (SCC), two were adenocarcinomas, and one case was small cell lung carcinoma. All cases involving SCC carcinomas and the case of small cell lung carcinoma presented skin metastases as the first sign of disease (Figure 5).

All SCC cases were positive for marker p40. One of the adenocarcinomas expressed TTF1, but the other tumor was negative, raising concerns for differential diagnosis (Figure 6). Histology, patient’s clinical and paraclinical data, and further studies confirmed lung adenocarcinoma. Small cell lung carcinoma was positive for CTK AE1/AE3, TTF1, and CD56.

Gastrointestinal tract metastases originated from the colorectum (three cases) and stomach (two cases). The most common site for metastasis in this category was the abdomen.

Of the two gastric tumors, one was adenocarcinoma (positive for CK7) and the other was gastrointestinal stromal tumor (GIST, positive for CD117). GIST is the only metastasis with mesenchymal origin identified in our study (Figure 7).

The cutaneous metastases from colorectal carcinoma were adenocarcinomas and expressed both CK20 and CDX2 (Figure 7). All patients in this category had a well-known history of the primary disease.

The five metastases from the urinary tract originated in the kidney. Histology revealed them as clear cell carcinomas, characterized by nests and groups of cells with clear cytoplasm and prominent nuclei and nucleoli, separated by thin connective tissue septa, presenting high vascularization. The most common site for skin metastases in this group was the scalp (two cases). All cutaneous metastases showed strong immunoreactivity for CD10 and PAX8 (Figure 8).

We encountered two cutaneous metastases originating in the reproductive tract. One was revealed as ovarian serous carcinoma (Figure 9), and the other was endometrial endometrioid carcinoma (Figure 10). Both patients had a well-documented pathological history. The metastasis of serous ovarian carcinoma presented a solid and papillary architecture, with papillary branching and cribriform areas, and the tumor cells were cuboidal or columnar, with marked atypia. The tumor expressed ER. The endometrioid carcinoma presented groups of atypical cells located in the upper dermis, which expressed CTK AE1/AE3 and PAX8.

## 4. Discussion

Due to the rarity of cutaneous metastases from internal organ malignancies, with the most relevant reference of 0.2% according to the study conducted by Vernemmen et al. in 2022 [4,5,6,7,9], their recognition can present a diagnostic challenge, especially when the primary tumor is not diagnosed and the patient has no known pathological history. The distinct histological features require immunohistochemistry confirmation and the choice of diagnostic markers requires careful examination and an experienced pathologist. In our study, only around 0.5% of skin specimens evaluated in our hospital represented metastases of internal organs malignancies [9].

Cutaneous metastasis was the initial indication of a clinically primary malignancy in seven of our patients, with an incidence two to six times higher than reported in other similar studies [18,19,20,21]. The lung was the primary tumor site in most of them, and the cutaneous metastasis was in the proximity of the tumor, predominantly in the anterior chest wall, the upper extremities, and the abdomen. The other two cases involved breast carcinoma.

In lung cancer, metastasis to the skin is rare compared to other organs such as the brain, bones, and liver, and according to medical studies, it is observed in around 1–12% of advanced lung cancer. They represent a significant indicator of advanced disease associated with poor prognosis, with an average survival ranging between three and six months in most studies [13,18,19]. All histological subtypes of lung cancer may secondarily involve the skin.

Four lung cancers were squamous cell carcinoma; thus, the main challenge was excluding a primary cutaneous neoplasm. The lack of epithelial involvement and the deep location of the tumor were important details for the diagnosis. The lung adenocarcinomas are positive for TTF1 and Napsin A. We usually use TTF1 marker to confirm the diagnoses. One case was positive, but the other one was negative for the marker, becoming a diagnostic challenge. According to the scientific literature, more than 50% of the cutaneous metastases of lung cancer are negative for this marker, due to the poorly differentiated nature of the adenocarcinoma in skin metastases [9,20].

Breast cancer often spreads to the anterior chest wall, while gastrointestinal cancers typically metastasize to the abdominal region. These locations support the hypothesis that the cutaneous metastases mostly result through regional dissemination, likely due to the lymphatic spread of the cancer cells. Some cutaneous metastases appear in more distant areas. For instance, the scalp is a common site for metastases from clear renal cell carcinoma. This is potentially due to the scalp’s high vascularity, making it a frequent site for cutaneous metastases from various types of cancers [17,21].

Breast cancer is known to be the second most frequent source of cutaneous metastases after melanoma, with an incidence ranging from 23% to 34% in patients with metastatic disease [22]. The opposite occurs in ovarian carcinoma, where cutaneous involvement is rare and is observed in less than 4% of cases [23]. Our study results also highlight that breast cancer is the primary tumor to metastasize to the skin in women, while ovarian cancer represents the least frequent one. The higher incidence of cutaneous metastases in breast cancer can be attributed to several factors. The anatomical proximity of the breast to the skin ensures direct extension and lymphatic spread to the dermis and subcutis [24]. Also, breast cancer has a predilection for spreading to various distant sites, including the skin, due to its aggressive nature and common pathways of metastatic spread [25,26,27]. Ovarian carcinoma typically disseminates to the peritoneum and pleura, with cutaneous metastasis being a late and rare event [28,29,30,31]. The mechanisms underlying this rare occurrence are not well understood, but it is thought to be related to the hematogenous or lymphatic spread, and in some cases, with direct invasion from contiguous structures [32,33]. The rarity of skin involvement in ovarian cancer is not proportional with its aggressive progression [34,35]. The most important differential diagnoses in these cases are primary adenocarcinomas of the skin, which originate in the adnexal structures. Immunohistochemistry is essential for determining the origin of the tumor. For breast, markers ER, PR, GATA3, and mammaglobin are extremely useful. For ovarian carcinoma, ER, WT1, p53, and PAX8 are relevant [36,37,38].

Metastases to the skin from urinary system cancers such as renal cell carcinoma (RCC) and urothelial carcinoma are very rare, with an incidence of less than 3%. However, when they do occur, they are indicative of advanced disease and carry a poor prognosis [16,39,40].

Renal cell carcinoma is the most common urinary tract neoplasm to metastasize to the skin. These metastases usually present as solitary or multiple nodules, which can be red, blue, or flesh-colored and are often found on the abdomen, chest, or scalp. The underlying disseminating mechanism involves hematogenous spread, as RCC is known for its high vascularity and tendency for vascular invasion and blood dissemination [41,42].

Bladder cancer (urothelial carcinoma) can also metastasize to the skin, though this is extremely rare. The clinical presentation can vary, but common manifestations include the appearance of nodular lesions, plaques, or ulcerative lesions, often located near the primary tumor site, such as the lower abdomen or perineal region [43]. The dissemination is typically through direct extension, lymphatic, or hematogenous spread.

Metastases from the upper gastrointestinal tract are a rare occurrence. Umbilical metastasis is common for gastric adenocarcinomas (Sister Mary Joseph’s nodule), but other areas of the torso can also be involved [44]. Colorectal cancer causes metastases to the skin in less than 5% of cases, according to some studies. The most common sites of dissemination for this type of cancer are the liver and the lungs. Therefore, metastasis to the skin is an event associated with poor prognosis, regardless of the skin area involved [45,46].

Gastrointestinal stromal tumor is the only mesenchymal metastasis identified in our study. Cutaneous metastasis from GISTs is exceedingly rare. These entities are the most common mesenchymal tumors of the gastrointestinal tract, the majority of which are in the gastric wall. Most metastatic GIST tumors are located in the abdomen or the liver, while extra-abdominal dissemination is very rarely encountered. The most important parameters that highlight the malignant potential of GIST are given by size, mitotic index, and location [47,48] ].

Visceral metastases on the skin are, overall, a rare encounter, with percentages varying from study to study. The diagnosis of these tumors is crucial for the therapeutic management of the patients affected. Differential diagnosis and finding the origin of the tumor are the most important histological challenges in daily practice, particularly in those cases where the primary tumor has not yet been identified.

## 5. Conclusions

Cutaneous metastases represent an important diagnostic pitfall in dermatopathology. Their identification can be difficult, as they may be the first presentation of an underlying cancer. Our study highlights the morphological aspect and immunohistochemical profile of the cutaneous metastases diagnosed in our center. We highlight the frequent primary sites, areas more commonly affected, differential diagnoses, and helpful markers. All these reports are important, given the poor prognosis associated with cutaneous metastases, and early recognition is essential in optimizing patient care.

## Figures and Tables

**Figure 1 dermatopathology-12-00014-f001:**
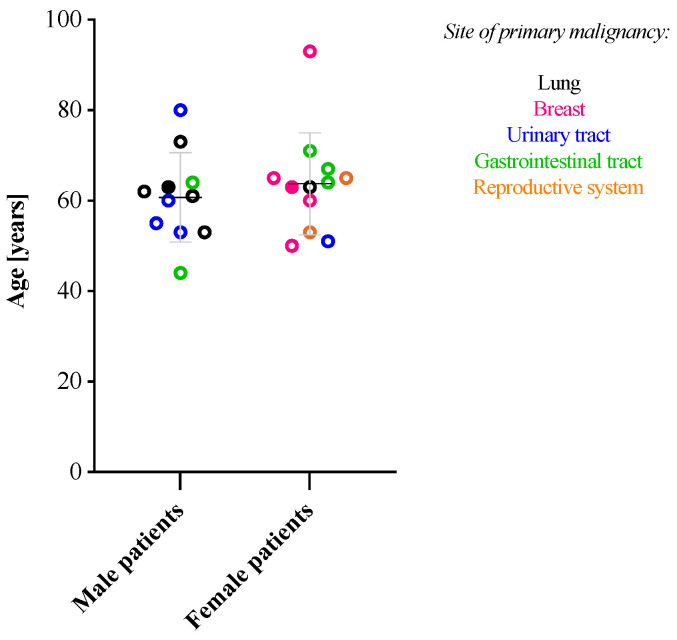
Age distribution of the patients included in the study, divided into males and females. The color-coding was used to mark the localization of the primary tumor (lung—black, breast—pink, urinary tract—blue, gastrointestinal tract—green, and reproductive system—orange), and solid circles were used to highlight the patients with multiple cutaneous metastases.

**Figure 2 dermatopathology-12-00014-f002:**
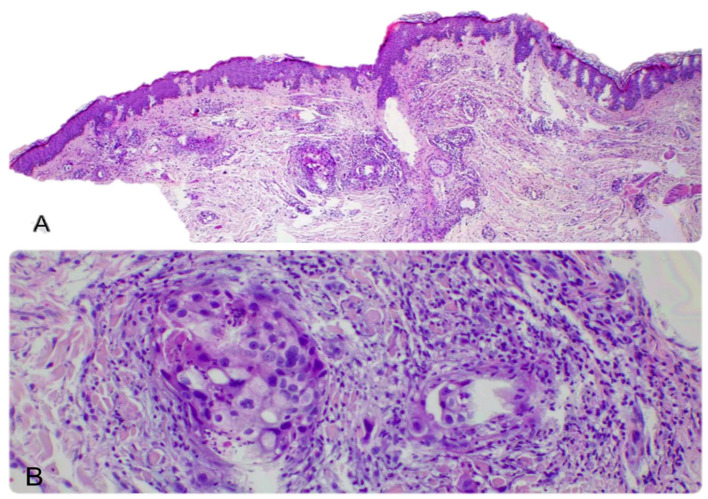
Cutaneous metastases localized in the dermis and subcutis with the following primary tumor: (**A**) invasive breast carcinoma of no special type (NST), 50× magnification; (**B**) invasive breast carcinoma of no special type (NST), with tumor cells distributed in nests, showing high pleomorphism, pale eosinophilic cytoplasm, hyperchromatic nuclei, and at 200× magnification.

**Figure 3 dermatopathology-12-00014-f003:**
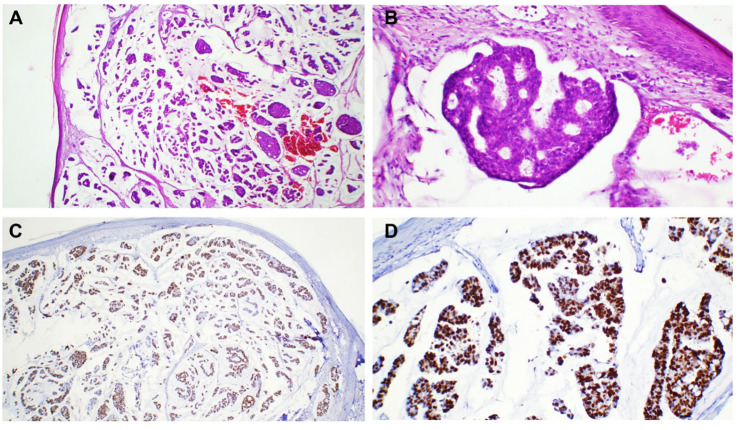
Cutaneous metastases localized in the dermis and subcutis with the following primary tumor: (**A**) invasive mucinous breast carcinoma, 50× magnification; (**B**) invasive mucinous breast carcinoma, HE, 200× magnification; and (**C**,**D**) clusters of tumor cells with uniform positive reaction marking the estrogen receptor in tumor nuclei, 50× magnification (**C**) and 200× magnification (**D**), respectively.

**Figure 4 dermatopathology-12-00014-f004:**
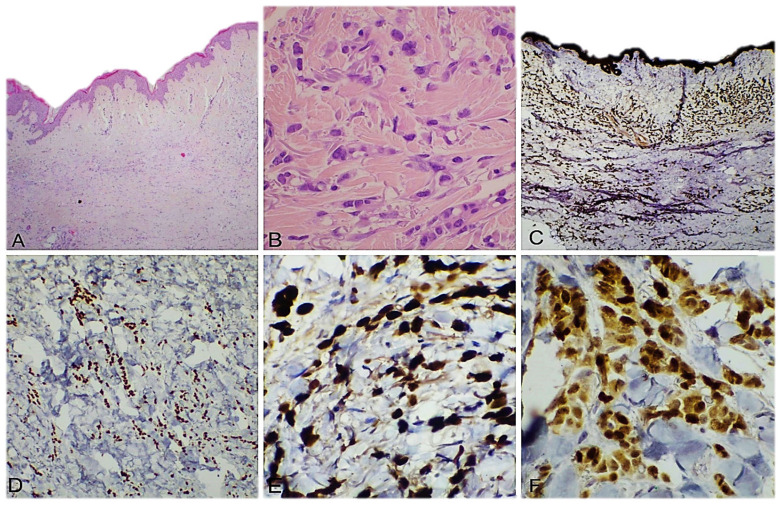
Cutaneous metastases localized in the dermis and subcutis with the following primary tumor: (**A**) invasive lobular carcinoma, we observe “crowded dermis”, 50× magnification; (**B**) invasive lobular carcinoma, tumor cells arranged in chords or isolated, HE, 200× magnification; (**C**) tumor cells with uniform positive reaction marking CTK AE1/AE3 cells, 50× magnification; (**D**) tumor cells with uniform positive reaction marking GATA3, 100× magnification; (**E**) tumor cells with uniform positive reaction marking ER, 40× magnification; and (**F**) tumor cells with uniform positive reaction marking PR, 400× magnification.

**Figure 5 dermatopathology-12-00014-f005:**
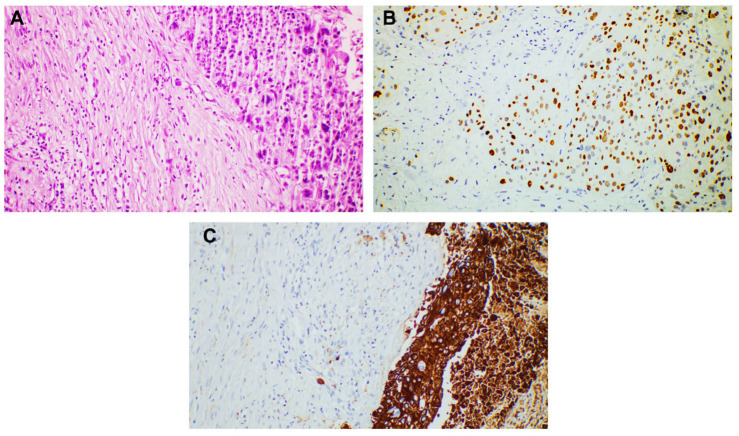
Cutaneous metastasis of lung carcinoma. (**A**) Invasive lung squamous cell carcinoma, 100× magnification, HE; (**B**) invasive lung squamous cell carcinoma, immunohistochemistry reaction with p40, 100× magnification; and (**C**) invasive lung adenocarcinoma, immunohistochemistry reaction with CTK AE1/AE3, 100× magnification.

**Figure 6 dermatopathology-12-00014-f006:**
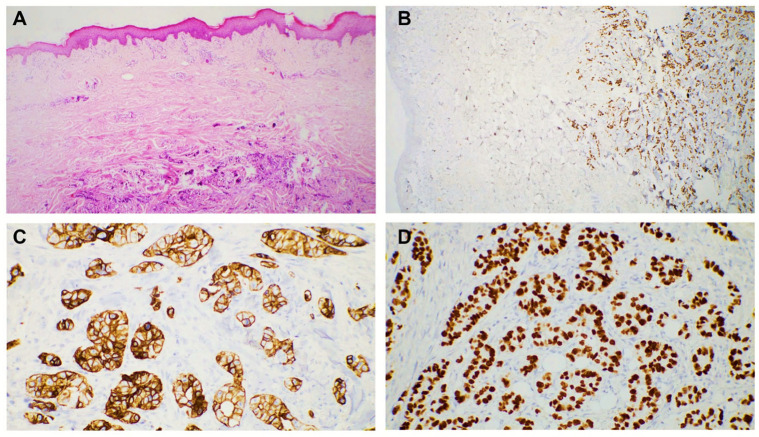
Cutaneous metastases of lung carcinoma. (**A**) Invasive lung adenocarcinoma, 50× magnification, HE; (**B**) invasive lung adenocarcinoma, immunohistochemistry reaction with TTF1, 100× magnification; (**C**) invasive lung adenocarcinoma, immunohistochemistry reaction with CTK AE1/AE3, 200× magnification; and (**D**) invasive lung adenocarcinoma, immunohistochemistry reaction with TTF1, 200× magnification.

**Figure 7 dermatopathology-12-00014-f007:**
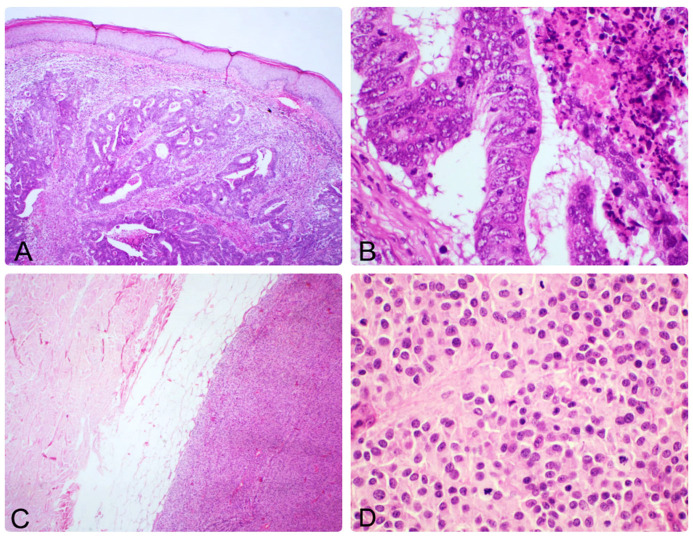
(**A**) Skin metastasis of colonic adenocarcinoma; neoplastic lesion in the dermis with extension towards subcutaneous tissue. Glandular differentiation and necrosis are evident, 50× magnification. (**B**) Cytologic details of metastatic colonic adenocarcinoma are observed, such as loss of polarity, enlarged nuclei with stratified aspect, mitoses, and areas of necrosis. 400× magnification. (**C**) GIST-Metastasis involving the hypodermis is showing a solid tumor proliferation with pushing borders, 50× magnification. (**D**) Tumor cells present round or ovoidal shape, eosinophilic cytoplasm, and hyperchromatic nuclei, with increased mitotic activity, 400× magnification.

**Figure 8 dermatopathology-12-00014-f008:**
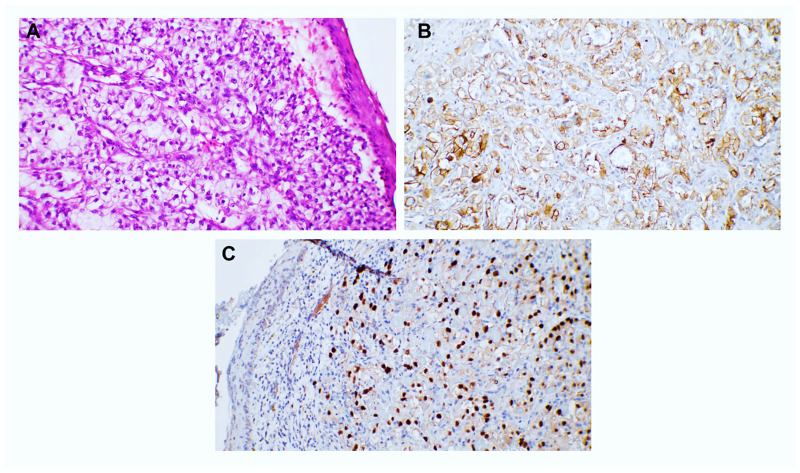
(**A**) Skin metastasis of renal cell carcinoma, with tumoral lesion in the dermis and subcutaneous cellular tissue, with ulcerated areas, foci of inflammatory infiltrate. Rows, sheets, and tubular structures containing cells with abundant clear and eosinophilic cytoplasm, vesicular chromatin nuclei with prominent nucleolus, and frequent mitotic figures were observed, 5× magnification. (**B**) Immunohistochemical staining with CD10, positive membranous expression, 20× magnification. (**C**) Immunohistochemical staining with PAX8, positive nuclear expression, 5× magnification.

**Figure 9 dermatopathology-12-00014-f009:**
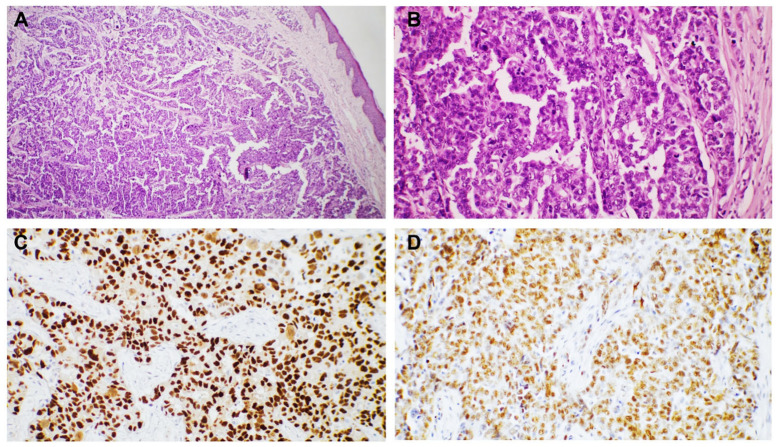
(**A**,**B**). Skin metastasis of serous cell carcinoma, seen as a tumor proliferation in the dermis and subcutaneous tissue. Rows, sheets, and papillary structures containing cells with eosinophilic cytoplasm, hyperchromatic nuclei with prominent nucleolus, and frequent mitotic figures were observed, 50× magnification and 100× magnification, respectively. (**C**,**D**) Immunohistochemical staining with ER, positive nuclear expression, 5× magnification and 100× magnification, respectively.

**Figure 10 dermatopathology-12-00014-f010:**
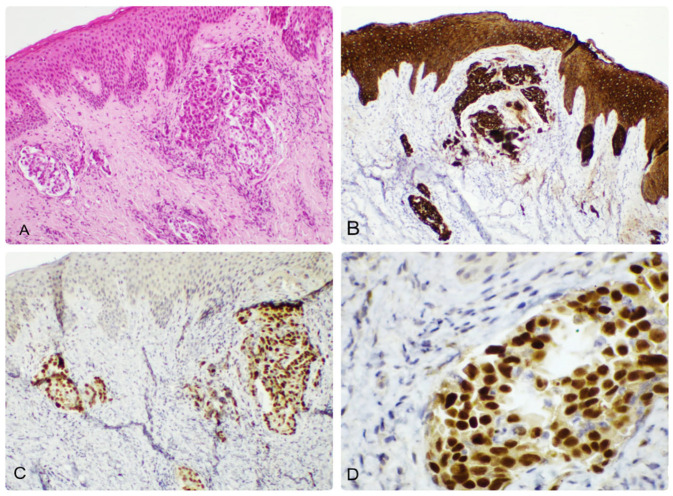
(**A**) Skin metastasis of endometrial endometrioid carcinoma, seen as a tumor proliferation in the dermis. Plaques of tumor cells with infiltrative pattern and marked atypia are observed, along with stromal retraction, 50× magnification. (**B**,**C**). Immunohistochemical staining with CTK AE1/AE3, positive cytoplasm expression, 50× magnification and 100× magnification, respectively; (**D**) Immunohistochemical staining with PAX8, positive nuclear expression, 200× magnification.

**Table 1 dermatopathology-12-00014-t001:** Panel of antibodies.

Estrogen (ER)	anti-Estrogen Receptor (ER) (SP1) Rabbit Monclonal Primary Antibody, VENTANA
Progesteron (PR)	anti-Progesterone Receptor (PR) (1E2) Rabbit Monoclonal Primary Antibody),VENTANA
GATA3	GATA3 (L50-823) Mouse Monoclonal Primary Antibody, VENTANA
E-Cadherin	E-Cadherin (EP700Y), Rabbit Monoclonal Antibody, CELL MARQUE
CTK AE1/AE3	(PCK26),Anti-Pan Keratin (AE1/AE3/PCK26) Primary Antibody), VENTANA
Cytokeratin 7 (CK7)	(SP52), Rabbit Monclonal Antibody)
Cytokeratin 20 (CK20)	(SP33), Rabbit Monclonal Primary Antibody), VENTANA
CDX2	EPR2764Y)) Rabbit Monoclonal Antibody),VENTANA
TTF1	SP141, Rabbit Monoclonal Antibody) VENTANA
p40	anti- p40 (BC28) Mouse Monoclonal Primary Antibody,), VENTANA
CD56	anti-CD56 (123C3) Mouse Monoclonal Primary Antibody, VENTANA
CD10	anti-CD10 (SP67) Rabbit Monclonal Primary Antibody, VENTANA
CD 117	PATHWAY Anti-c-KIT (9.7) Primary Antibody, VENTANA
PAX-8	Anti-PAX 8, anti-PAX 8 (MRQ-50) Mouse Monoclonal Antibody, CELL MARQUE

**Table 2 dermatopathology-12-00014-t002:** Cutaneous metastases and their primary origin.

Site of Primary Malignancy	Number of Cutaneous Metastases						
	Chest	Trunk	Scalp	Shoulder	Abdomen	Upper Extremity	Total
Breast	4	0	0	0	0	2	6
Lung	3	1	0	1	0	2	7
Gastrointestinal tract	0	1	0	1	2	1	5
Urinary tract	1	0	2	0	1	1	5
Reproductive tract	0	1	0	0	1	0	2
Total	8	3	2	2	4	6	25

## Data Availability

The data presented in this study are available at the request of the corresponding author for privacy and ethical reasons.

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
