# Peer review of "Cutaneous Metastases—Histological Particularities of Multifaceted Entities"

_dermatopathology, 2025, doi:10.3390/dermatopathology12020014_

Round 1
Reviewer 1 Report
Comments and Suggestions for Authors
This is a well-planned and executed scientific paper that offers a comprehensive overview of skin metastases, focusing on the prevalence of the most common primary tumor sites and the cutaneous regions most frequently affected.
I enjoyed reading it.
Apart from that here are some minor proposed revisions:
line 54 would like to see some percentages
line68 it is Cytokeratin (fix the table) and insert model of the automatic stainer
line 71 what are the criteria
line 117 Invasive mucinous breast carcinoma of no special type pure Mucinous or NST mixed? (looks pure mucinous in photo)
Usually in microphotos the magnification (x) takes into consideration both eyepiece (or camera) and magnifying lens (multiply per 10 your values in all photos)
line 122 the diagnosis of a lobular carcinoma is validated by e-cadherin (if not mention history of lobular carcinoma)
line 137 Small cell lung carcinoma not small cell neuroendocrine carcinoma.
line 160 GIST rarely metastasizes to the skin. if you have such case you should publish it. were the metastases compatible with the so called "Sister Mary Joseph nodule"
line 209 in the discussion paragraph you do not mention anything about GI (gastrointestinal)
Author Response
Dear reviewer, thank you very much for your response and comments, we appreciate it a lot!
Regarding your suggestions,
line 54 would like to see some percentages- we did, hopefully it is what you had in mind as well,
line68 it is Cytokeratin (fix the table) and insert model of the automatic stainer-we did both, thank you for correcting,
line 71 what are the criteria-we added it in the paragraph,
line 117 Invasive mucinous breast carcinoma of no special type pure Mucinous or NST mixed? (looks pure mucinous in photo)- correct, the primary tumor was mixed (the specimen was not analyzed in our hospital, we just found it in the patient's history) while the metastasis is mucinous, we corrected that and kept it solely as mucinous,
Usually in microphotos the magnification (x) takes into consideration both eyepiece (or camera) and magnifying lens (multiply per 10 your values in all photos)- thank you for this comment, we did for all figures,
line 122 the diagnosis of a lobular carcinoma is validated by e-cadherin (if not mention history of lobular carcinoma)- we found about the patient's history after the IHC protocol was on, we also performed E-cadherin and it was negative, we added it in the results,
line 137 Small cell lung carcinoma not small cell neuroendocrine carcinoma- correct, thank you, translation problem,
line 160 GIST rarely metastasizes to the skin. if you have such case you should publish it. were the metastases compatible with the so called "Sister Mary Joseph nodule"- yes it is, we are preparing another manuscript with details regarding the case,
line 209 in the discussion paragraph you do not mention anything about GI (gastrointestinal)- we added it in the discussions.
All changes are highlighted in the manuscript. Thank you very much for your time, dedication and all your comments!
Reviewer 2 Report
Comments and Suggestions for Authors
The work describes the morphological aspect and the immunohistochemical profile of skin metastases from internal organ cancer, based on the experience collected. The interest of these localization concerns the possible pitfall that they represent, mainly due to the rarity with which they manifest.
This is a linear and traditional histopathological diagnostic work that adds case history and maintains the memory of a critical diagnostic problem.
The data reported are exhaustive in all their aspects.
The iconography is complete, exhaustive and well represented.
The first 40 lines can be shortened because they are very analytical and not strictly relevant to the topic discussed. They can disorient the reader who is focused on skin metastases by the title.
A case of GIST metastasis is reported in line 160, but it is not shown or further discussed. Since this is the only metastasis from a mesenchymal tumor, I think it is appropriate to decide whether to treat it or not. Therefore, if affirmative, it should also be reported in the iconographic part and in the discussion. Alternatively, I think it is appropriate to exclude it, limiting ourselves to only skin metastases of epithelial malignant tumors.
I believe that with a revision of the introduction and a decision on a GIST case just mentioned, the work deserves to be published.
Author Response
Dear reviewer, thank you very much for your comments!
As you suggested, we shortened the introduction and eliminated the aspects that were describing general diseases, hopefully the text is clearer now.
We added a paragraph that mentions the GI tract metastasis as well, and added a few words in the results section, too. GIST in the skin is a very rare encounter indeed, and we are planning on publishing the case separately. We thought about the idea of eliminating it from the article since it is indeed the only mesenchymal tumor, but eventually we decided to keep it.
Thank you for your time and review, we appreciate it a lot!